# Ultrasound Assessment of Sarcopenia in Alcoholic Liver Disease

**DOI:** 10.3390/diagnostics14171891

**Published:** 2024-08-28

**Authors:** Vlad-Teodor Enciu, Priscila Madalina Ologeanu, Carmen Fierbinteanu-Braticevici

**Affiliations:** 1Internal Medicine II and Gastroenterology Department, Emergency University Hospital Bucharest, Carol Davila University of Medicine and Pharmacy, 050474 Bucharest, Romania; priscila-madalina.ologeanu@drd.umfcd.ro (P.M.O.); carmen.fierbinteanu@umfcd.ro (C.F.-B.); 2Emergency University Hospital, 050098 Bucharest, Romania

**Keywords:** ALD, sarcopenia, ultrasound

## Abstract

Malnutrition frequently affects patients with alcoholic liver disease (ALD), with important impacts on disease prognosis. Sarcopenia, the clinical phenotype of malnutrition characterized by skeletal muscle loss, is the major component responsible for adverse events in this population. The aim of this study is to assess the use of ultrasound (US) skeletal muscle performance in stratifying ALD disease severity. We recruited 43 patients with ALD and divided them into two groups: alcoholic hepatitis (AH) and alcoholic cirrhosis (AC). We evaluated disease-specific clinical and biological parameters and their relation to US Rectus Femoris muscle (RFM) measurements, including RFM thickness, stiffness (RFMS) and echogenicity (RFE). A thirty-seconds chairs stand test (30sCST) was used as the sarcopenia surrogate test. RMF thickness correlated with platelet count and serum albumin (*p* < 0.001). Both RFM and RFMS correlated with disease severity (*p* < 0.001) and 30sCST (*p* < 0.001, *p* = 0.002). Patients with AH had more severe US muscle abnormalities compared to AC (RFMS 1.78 m/s vs. 1.35 m/s, *p* = 0.001) and the highest prevalence of RFE (χ^2^ = 8.652, *p* = 0.003). Rectus Femoris US assessment could represent a reliable tool in the diagnosis and severity stratification of ALD-induced sarcopenia.

## 1. Introduction

Malnutrition is one of the most significant comorbidities in alcoholic liver disease (ALD), with an important impact on both morbidity and mortality [1]. However, malnutrition is a general term defined as a disorder of inadequate intake or uptake of nutrients leading to a decrease in total body cell mass. Sarcopenia, the clinical phenotype of malnutrition with prognostic importance in ALD, characterized by skeletal muscle loss, has been proven to be a major component responsible for adverse outcomes in this population category [2]. Both alcohol intake and impaired function of the liver contribute to the development of sarcopenia [3].

Sarcopenia is mainly assessed clinically using a combination of anthropometric measurements and other dynamic tests [4]. However, these tests do not measure skeletal muscle mass directly and are not applicable in patients with decompensated ALD, since most of the clinical parameters are influenced by oedema or ascites [5]. Moreover, dynamic tests such as grip strength cannot differentiate between sarcopenia and frailty, a condition that is either attributed to age-related physiological decline or to disease-induced impaired muscle contraction and weakness [6]. Cross-sectional imaging using computer tomography (CT) scans is considered the gold standard for the precise quantification of sarcopenia, but is irradiating, time-consuming, and expensive to implement as a suitable screening tool [7,8,9]. Thus, other non-invasive, point-of-care methods are analyzed for an accurate assessment of sarcopenia allowing prompt disease stratification and therapeutic intervention [7].

The ultrasound (US) measurement of muscle abnormalities is a promising, readily available and reproducible tool used in assessing sarcopenia [10]. This method has already been validated in geriatric [11] and intensive care unit patients [12]. During a consensus conference, a standardization of US sarcopenia assessment was proposed [13]. Five muscle parameters were illustrated for the ultrasound characterization of sarcopenia, specifically muscle thickness, muscle cross-section area, pennation angle, fascicle length and echogenicity. Microcirculation damage and impaired nitric oxide synthesis have also been proposed as pathophysiological mechanisms of sarcopenia development through alterations of cellular metabolism, mitochondrial abnormalities and ultimately apoptosis. Glucose homeostasis is tightly linked to microcirculation, thus capillary rarefaction may contribute to insulin resistance through reduced glucose uptake [14,15]. This theory could also explain the increased muscular deposits of adipose tissue and the development of myosteatosis [16]. The advancements in US techniques that utilize microvascular imaging and contrast-enhanced ultrasound (CEUS) make muscle microvasculature quantifiable [17]. Skeletal muscle stiffness is another researched parameter analyzed through share-wave elastography (SWE) that was applied in some pilot studies [18,19,20]. The results are controversial, correlating a decrease in muscle stiffness with muscle weakness [18,19], or on the contrary, an increase in muscle stiffness with frailty and disease activity [20].

Taking into account the lack of non-invasive, point-of-care methods for the assessment of sarcopenia, our study sought to investigate the utility of skeletal muscle ultrasound alteration in the diagnosis and stratification of ALD-induced sarcopenia as a novel application.

## 2. Materials and Methods

### 2.1. Study Design and Inclusion Criteria

This is a prospective pilot study on patients diagnosed with ALD between December 2023 and May 2024. We included patients with alcoholic hepatitis (AH) and alcoholic cirrhosis (AC), divided into 2 groups (group 1—AH, group 2—AC) and a control group. The control group consisted of healthy patients with no major comorbidities that could influence either muscle functional tests or ultrasound measurements. AH diagnosis was established following the National Institute of Alcohol Abuse and Alcoholism (NIAAA) criteria for probable AH, which includes: history of heavy chronic alcohol consumption (>50–60 g/day for men and >40 g/day for women), recent onset of jaundice (<8 weeks), total serum bilirubin >3 mg/dL and characteristic de Ritis ratio AST:ALT > 1.5, with AST not exceeding 600 U/L. We excluded other hepatic etiologies through screening for viral hepatitis, hemochromatosis and Wilson’s disease, and a thorough assessment of patient’s prescription for known liver toxic drugs. Also, patients with AH and clinical signs of ethanol intoxication, ethylic tremor or hepatic encephalopathy grade ≥ II were excluded as this would influence the functional tests for sarcopenia.

Patients with alcoholic cirrhosis were previously diagnosed based on clinical, biologic and ultrasound modifications. We excluded CHILD C decompensated cirrhosis with large-volume ascites (clinically detectable) or oedema that could influence ultrasound measurements.

Following the recommendations of EWGSOP2, we chose the 30 s chair-stand test (30sCST) as the clinical surrogate muscle functional test for probable sarcopenia. Each patient was asked to perform stand-ups from a folding chair without arms, with the hands crossed and held against their chest. The number of successful and correct repetitions within a timespan of 30 s was recorded. The 30sCST was compared between all groups.

Patients were clinically examined and blood samples were obtained on admission. The control group was assessed during a routine check-up that included clinical examination, blood samples and muscle ultrasound measurements. Weight and height were measured and body mass index (BMI as kg/m^2^) was calculated. Patients with BMI > 30 kg/m^2^ were excluded as this might additionally alter the 30sCST outside of its intended use. Patients with fever were screened for infections using chest X-ray, urinary analysis and blood cultures. We performed ascites neutrophil counts in patients with ascites and abdominal pain for suspected spontaneous bacterial peritonitis (SBP). Patients with symptomatic infections and renal impairment were also excluded from the study.

### 2.2. Ultrasound Assessment of Rectus Femoris

The skeletal muscle ultrasound assessment was conducted following the latest SARCUS recommendations [13]. Ultrasonographic measurements were obtained with a B-mode linear transducer (5.0–7.5 MHz) using Acuson S2000 (Siemens AG, 91052 Erlangen, Germany). Patients were assessed in supine position, after resting for at least 5 min, with the lower limbs extended and knees relaxed. The right Rectus Femoris muscle thickness (RFM) was measured via a transverse scan at the mid-part of the thigh by applying minimal pressure with the transducer. The antero-posterior diameter of the RF was measured as the distance between the superficial and deep fascias. Rectus Femoris echogenicity (RFE) was evaluated qualitatively using gray-scale analysis (Heckmatt scale) in comparison to the surrounding adipose tissue and bone (Figure 1) [21]. Rectus Femoris muscle stiffness (RFMS) was calculated via longitudinal scanning by Acoustic Radiation Force Impulse (ARFI) with five measurements and the results have been expressed as mean shear wave velocity (Vm m/s) and interquartile range (IQR) (Figure 1) [22,23]. The measurements were performed by a hepatologist with more than 20 years of experience in ultrasonography and elastography techniques. US skeletal muscle findings were compared between groups and with 30sCST.

### 2.3. Statistical Analysis

Microsoft Excel 2021 MSO (Version 2407 Build 16.0.17830.20166, Microsoft Corporation, Redmond, WA, USA) was used to compose the database and a statistical analysis was conducted using SPSS Statistics version 26 (IBM Corporation, Armonk, NY, USA).

Quantitative variables with normal distribution and equal variances were analyzed using *t*-test and one-way ANOVA. Qualitative variables and those without a normal distribution were assessed using Mann–Whitney U and Chi-Square through crosstabulation. The results have been expressed as mean and standard deviation (SD) for normally distributed variables and as median and IQR for non-normally distributed variables. Association between variables was evaluated with Pearson or Spearman correlation and a *p*-value < 0.05 was considered statistically significant.

The study was approved by the Local Ethics Committee (36439/16 June 2022) and a detailed informed consent form was signed by each patient.

## 3. Results

### 3.1. General Characteristics of Study Population

Forty-three patients with ALD were included in the study, and six control subjects. Twenty-eight patients were included with AH (*n* = 8, 28% women, *n* = 20, 72% men) and fifteen with alcoholic cirrhosis (*n* = 6 Child-Pugh Class A, *n* = 9 Child–Pugh Class B). More males were present in both groups (*n* = 20 and *n* = 10), with no significant difference in age (*p* = 0.19). Average BMI was similar in both groups (*p* = 0.44). Only AH patients had either low-volume ascites (*n* = 12) or grade I hepatic encephalopathy (*n* = 7). Hepatic failure and inflammation biomarkers were significantly higher in the AH group, as expected. Most of the AH patients had mild forms that did not require corticosteroid treatment (Median MDF = 28.5). Only two patients died at 30 days from a severe form of AH. Group characteristics are illustrated in Table 1. 

### 3.2. Clinical and Biological Results

AC performed better than AH at 30sCST (11.47 vs. 10.8) (Figure 2), but without attaining statistical significance (*p* = 0.38). Additionally, 30sCST was influenced by disease severity (MDF r= −0.616) and parameters of hepatic disfunction (INR r= −0.678, serum albumin r= 0.645). We did not find a correlation between serum creatinine levels and either 30sCST or RF ultrasound measurement.

### 3.3. Rectus Femoris Ultrasound Measurement Results

Both groups had reduced but similar RFM thicknesses (1.34 cm IQR: 1.12–1.61 vs. 1.24 cm IQR: 1.03–1.65) (Figure 3). Although BMI correlated positively with 30sCST (r = 0.587), we did not find a correlation with RFM (*p* = 0.09) or RFMS (*p* = 0.62). Both RFM and RFMS significantly correlated with 30sCST (r = 0.786, r = −0.444), suggesting its utility in the integrated diagnosis of sarcopenia. Compared to control subjects, both AH and AC groups had lower 30sCST (*t* = −9.51, *p* < 0.001 and *t* = −8.58, *p* < 0.001), with reduced RFM thickness (*t* = −5.54, *p* < 0.001 and *t* = −5.5 *p* < 0.001). Regarding muscle stiffness, the AC group had lower RFMS (Vm = 1.35 m/s, IQR = 1.22–1.45) than the AH group (Vm = 1.78, IQR = 1.45–2.49), *p* = 0.001. Interestingly, RFMS was not necessarily corelated with RFM thickness (*p* = 0.36). Both RMF and RFMS correlated with disease severity in AH (r = −0.483, *p* = 0.009 and r = 0.558 *p* = 0.002) (Table 2).

Post-hoc pairwise comparisons using the Bonferroni correction revealed that RFE was the most prevalent in AH (χ^2^ = 8.652, *p* = 0.003).

## 4. Discussion

In this pilot study, we aimed to address the reliability of rectus femoris ultrasound measurements in the diagnosis of ALD-induced sarcopenia. Our results suggest that RF thickness, RFMS and RFE are important non-invasive parameters that correlate with the 30sCST sarcopenia surrogate test and disease severity.

Sarcopenia is a well-known prognostic factor in chronic liver diseases, and recently, novel methods of diagnosis have been developed in order to offer a more feasible diagnostic approach. Although our study did not include a comparison with cross-sectional imaging techniques, ultrasound skeletal muscle measurements have been proven to be a reliable diagnostic alternative to the conventional CT scan gold-standard [5,20,24,25]. These studies validated ultrasound muscle measurements when compared to the CT scan skeletal muscle index (SMI). Moreover, muscle functional tests, specifically the 30sCST, which was used as a sarcopenia surrogate marker in our study, are endorsed by the EWGSOP group as reliable clinical parameters [7]. Our results are similar to those of other studies, wherein decompensated liver states had lower muscle function [26]. As already reported, BMI alone cannot be used as a precise marker of sarcopenia, since the results are influenced by oedema and ascites, hence the need for a more integrated approach. Lastly, although creatinine levels are known to be related to malnutrition and sarcopenia, our study did not find a correlation with muscle functional tests or ultrasound measurement, suggesting that low serum creatinine levels are not reliable for the diagnosis of sarcopenia. Our findings regarding ultrasound skeletal muscle alteration in ALD patients are similar to those of other studies [2,20,24,25,27,28]. We showed that patients with ALD have reduced RFM thickness (1.34 cm and 1.24 cm), indicative of sarcopenia when compared to 30sCST, even in patients with normal BMI. In the AH group, a reduced RFM thickness and a high RFMS indicated disease severity. These findings are consistent with the results found in the study of Becchetti et al. regarding decompensated cirrhosis [20]. Thus, RFM thickness and RFMS could be implemented as severity parameters in alcoholic hepatitis. Interestingly, we did not find a correlation between RFMS and RFM thickness, suggesting that normal muscle fiber composition is influenced by specific pathophysiological mechanisms that can precede muscle mass reduction [29].

On this note, myosteatosis and myofibrosis have been recognized in the literature to induce different degrees of muscle impairment through increased deposits of adipose tissue or extracellular matrix [30,31]. This phenomenon is not necessarily related to obesity and can also be found in patients with cirrhosis, in relation to a chronic inflammatory state [32]. These statements are in concordance with our results regarding RFMS and RFE parameters in AH. Out of the three groups, AH had the highest RFMS and the highest prevalence in RFE, suggesting that this disease category is the most predisposed to either myofibrosis or myosteatosis. Skeletal muscle qualitative modifications in ALD are complex and can be attributed to both disease activity and the toxic effect of alcohol. Alcohol consumption leads to muscle autophagy, decreased proteasome activity and insulin-like growth factor, while disease activity induces cellular and metabolic abnormalities, consisting of differentiations of muscle stem cells into adipocytes, insulin resistance, hyperammonemia, mitochondrial dysfunction and apoptosis [33,34]. Studies including muscle biopsies are warranted in order to further understand qualitative muscle alterations, their ultrasound expression and their relation to disease severity.

Taking into consideration that ALD patients undergo periodic ultrasonography surveillance, ultrasound skeletal muscle measurements could represent an easy, low-cost, readily available strategy to monitor incipient sarcopenic or myosteatotic modifications.

Our study has some limitations. First of all, it has a small sample size, and our protocol did not include CT scans for the comparison of RFMS and RFE with cross-sectional imaging parameters. However, 30sCST is an approved clinical surrogate of sarcopenia, and our intention was to use it specifically, as it targets the muscle group of interest included in our study. Secondly, there is no approved standardized ultrasound skeletal muscle measurement protocol, specifically for muscle stiffness or echogenicity, so the results can be heterogenous. However, we followed the latest SARCUS update recommendations, implementing similar and optimal methods in unison with most ultrasound skeletal muscle studies.

To the best of our knowledge, this is the first study to quantify the muscle stiffness of ALD patients through ARFI.

## 5. Conclusions

In conclusion, Rectus Femoris ultrasound measurements could be used as a non-invasive diagnostic and prognostic tool in ALD, whether it reflects underlying sarcopenia or myosteatosis. Larger studies and technique standardization are required in order to facilitate the US assessment of ALD-induced muscle changes.

## Figures and Tables

**Figure 1 diagnostics-14-01891-f001:**
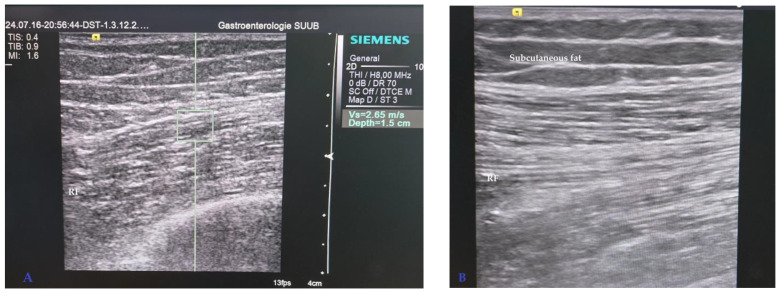
Ultrasound mid-thigh assessment of muscle stiffness (**A**) and muscle echogenicity (**B**) in a patient with ALD. Increased RFMS calculated by ARFI elastography expressed as mean shear wave velocity (Vs m/s) (**A**) and longitudinal ultrasound view of Rectus Femoris (RF) echogenicity with increased brightness compared to subcutaneous fat (**B**). RFMS: Rectus Femoris muscle stiffness. RF: Rectus Femoris. ALD: Alcoholic liver disease.

**Figure 2 diagnostics-14-01891-f002:**
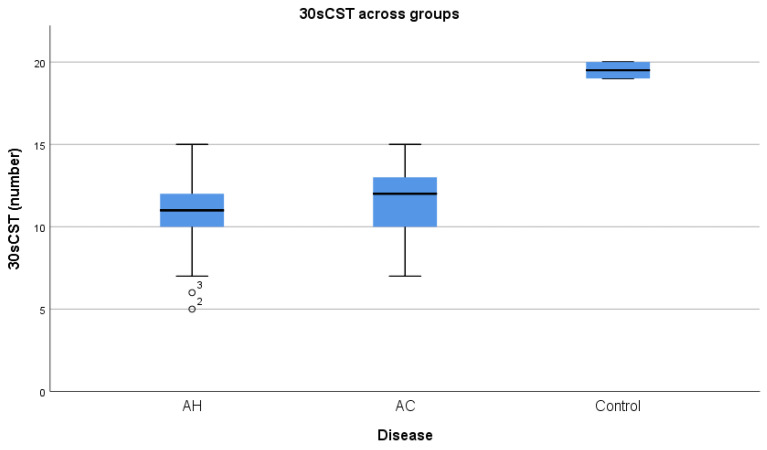
Median 30 s chair stand test (30sCST) distribution across groups, AH = alcoholic hepatitis, AC = alcoholic cirrhosis.

**Figure 3 diagnostics-14-01891-f003:**
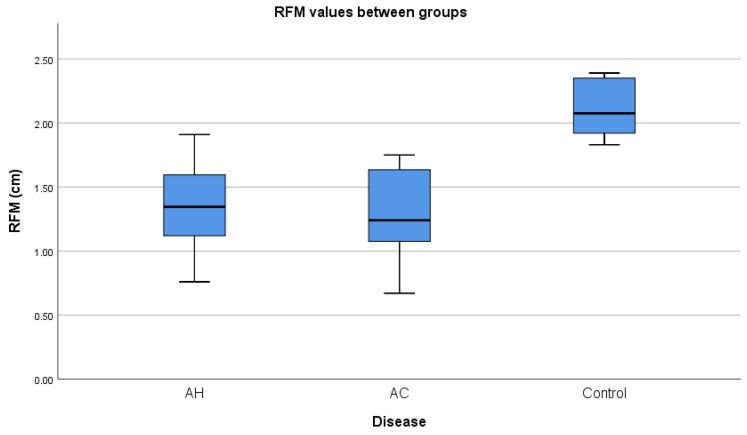
Median Rectus Femoris muscle thickness (RFM) distribution across groups. AH = alcoholic hepatitis, AC = alcoholic cirrhosis.

**Table 1 diagnostics-14-01891-t001:** Study group characteristics and comparison.

Variable	Alcoholic Hepatitis	Alcoholic Cirrhosis	Method	*p*-Value
Number	28	15	-	-
Sex (M/F)	20 M/8 F	10 M/5 F	Chi Square	0.746
Age (years)	53.89 (± 9.2)	50.13 (±8.4)	ANOVA	0.196
Ascites (0/1)	12 (43%)	0	Chi Square	0.003 *
Encephalopathy (0/1)	7 (25%)	0	Chi Square	0.034 *
Esophageal Varices (0/1)	7 (25%)	4 (26%)	Chi Square	0.905
BMI (kg/m^2^)	23.42 (±4.2)	22.36 (±3.5)	T test	0.440
WBC (×10^3^ cells/uL)	9.88 (±5)	9.86 (±3.3)	T test	0.988
PLT (×10^3^ cells/uL)	127.61 (±83)	164.6 (±53)	T test	0.126
INR	1.7 (±0.3)	1.53 (±0.3)	T test	0.006 *
ALT (U/L)	49.86 (±29.3)	58.33 (±22.7)	T test	0.336
AST (U/L)	134.11 (±71.4)	66.6 (±24.9)	T test	0.001 *
Creatinine (mg/dL)	0.59 (0.51–1.09)	0.87(0.7–1.1)	M-W U	0.161
Urea (mg/dL)	26 (17–55)	45 (41–54)	M-W U	0.044 *
Albumin (mg/dL)	2.75 (±0.77)	3.02 (±0.48)	T test	0.380
Total Bilirubin (mg/dL)	6.15 (4.67–12)	1.8 (1.4–2.1)	M-W U	0.001 *
ALP (mg/dL)	121 (94–169.5)	42 (41–55)	M-W U	0.001 *
GGT (U/L)	378 (122–768.75)	60 (55–110)	M-W U	0.001 *
CRP (mg/dL)	10.45 (3.15–26.5)	0.7 (0.6–2.5)	M-W U	0.001 *
LDH (U/L)	203 (170–250)	80 (77–89)	M-W U	0.001 *
Sodium (mEq/L)	131.43 (±7.32)	135.2 (±2.95)	T test	0.235
RFM (cm)	1.34 (1.12–1.61)	1.24 (1.03–1.65)	M-W U	0.760
RFMS (m/s)	1.78 (1.45–2.49)	1.35 (1.22–1.45)	M-W U	0.001 *
RFE (0/1)	22 (78.6%)	8 (55.3%)	Chi Square	0.086
30sCST (*n*)	10.8 (±2.1)	11.47(±2.23)	T test	0.380
MELD	17.07 (±7.24)	11.41 (±3.45)	T test	0.007 *
MDF	28.45 (25.2–58.44)	-	-	-
Child–Pugh	-	A—6 (40%) B—9 (60%)	-	-

Continuous normally distributed variables are illustrated as mean and SD = standard deviation. Continuous non-normally distributed variable are illustrated as median and IQR = interquartile range. Categorical variables are illustrated as frequency and percentage. BMI = body mass index, WBC = white blood cell count, PLT = platelet count, INR = international normalized ratio, ALT = alanine aminotransferase, AST = aspartate aminotransferase, ALP = alkaline phosphatase, GGT = gamma glutamyl transpeptidase, CRP= C reactive protein, LDH = lactate dehydrogenase, RFM = Rectus Femoris muscle thickness, RFMS = Rectus Femoris muscle stiffness, RFE= Rectus Femoris echogenicity, 30sCST = 30 s chair stand test, MELD = Model for End-Stage Liver Disease, MDF = Maddrey’s discriminant function, ANOVA = one-way analysis of variance, T test = independent samples T test, M-W U = Mann–Whitney U test, * = statistically significant at the 0.05 level.

**Table 2 diagnostics-14-01891-t002:** Bivariate correlations between RFM, RFMS and continuous variables.

Variable, r (*p*-Value)	RFM	RFMS
Age (years)	−0.178 (0.221)	0.021 (0.887)
BMI (kg/m^2^)	0.320 (0.090)	−0.025 (0.620)
PLT (×10^3^)	**0.512 (<0.001) ***	−0.034 (0.817)
INR	−0.571 (<0.001) *	**0.309 (0.031) ***
Creatinine (mg/dL)	0.217 (0.135)	0.135 (0.355)
Albumin (g/dL)	**0.549 (*p* < 0.001) ***	−0.80 (0.585)
RFM	-	0.067 (0.647)
RFMS	0.067 (0.647)	-
30sCST	**0.786 (*p* < 0.001) ***	**−0.444 (0.002)**
MDF	**−0.514 (*p* < 0.001) ***	**0.480 (<0.001) ***

BMI—body mass index; PLT—platelet count; INR—international normalized ratio; RFM—Rectus Femoris thickness; RFMS—Rectus Femoris stiffness; 30sCST—30 s chair stand test; MDF—Maddrey’s discriminant function, * = statistically significant at the 0.05 level.

## Data Availability

The data presented in this study are available on request from the corresponding author. The data are not publicly available due to privacy and ethnical restrictions.

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
