# Peer review of "Ultrasound Assessment of Sarcopenia in Alcoholic Liver Disease"

_diagnostics, 2024, doi:10.3390/diagnostics14171891_

Round 1

Reviewer 1 Report

Comments and Suggestions for Authors

FIRST of all, except the limitations mentioned by the Authors themselves I have some reservations:                              
- control group consisited of What kind of patients?

- in „material and method” section shouldn’t be any explanation of the results (move it to discussion)                       - Authors didn’t really assess the acrivity of the disease as no patients had biopsy done (this study didn’t include histopathological state of livers which in my opinion is serious disadvantage)

- problems in ALD are Rather complex and patients often have coexisting disorders which may influence the results thus I Think that their conclusions/ especially with this group sample/ are a little exaggerated

Comments on the Quality of English Language

Minor edutorial revision

Author Response

Comment 1. Control group consisted of what kind of patients?

Response 1. Control group included healthy patients with no major comorbidities that could influence either 30CST or ultrasound muscle measurements.

Comment 2. In “Material and Method” section there shouldn’t be any explanations of the results (move it to discussion).

Response 2. Demographics were removed from material and methods accordingly.

Comment 3. Authors didn’t really assess the activity of the disease as no patients had biopsy done (this study didn’t include histopathological state of livers which in my opinion is serious disadvantage).

Response 3. Steatohepatitis is a diagnosis based on liver histology, while alcoholic hepatitis is a clinical diagnosis and liver histology is not mandatory, especially when patients already have the diagnosis of cirrhosis.  We agree that gold standard for disease activity is histopathological assessment, but this is rarely feasible in acute decompensation, specifically in alcoholic hepatitis where the risk of bleeding is high and the only option remains transjugular guided biopsy. Diagnosis of “probable” alcoholic hepatitis is now made using NIAAA criteria with very good accuracy. The stratification of the disease severity is based on widely used and validated severity scores. We presented results related to disease severity through consecrated severity scores and biological parameters highlighting their relation to ultrasound muscle alterations. This study did not aim to assess histopathological liver modifications and their relation to sarcopenia.

Comment 4. Problems in ALD are rather complex and patients often have coexisting disorders which may influence the results; thus, I think that their conclusions/ especially with this group sample/ are a little exaggerated.

Response 4. Indeed, ALD is rather complex and results may be influenced by other factors. We know the impact of alcohol in heart disease and different types of cancer, which can independently influence muscle function. This is why we excluded the patients with those comorbidities. Moreover, we excluded patients with clinically evident muscle function impairment due to ethanol intoxication, tremor or hepatic encephalopathy grad ≥ II, so the functional test would be relevant. Nonetheless, imagistic muscle alterations have been studied both in ALD and non-ALD and their utility in disease severity is warranted. (https://www.journal-of-hepatology.eu/article/S0168-8278(21)02174-7/fulltext)

As mentioned, ultrasound muscle assessment of sarcopenia is a rather novel approach that requires standardization and larger studies. This pilot study was not intended to represent a cornerstone in sarcopenia assessment, but rather to pave the way for further research. Taking this into consideration, we made the appropriate text modifications in the “Conclusion” section.

Reviewer 2 Report

Comments and Suggestions for Authors

This paper investigates the utility of ultrasound measurements in assessing sarcopenia among patients with alcoholic liver disease (ALD). The study focuses on using ultrasound to measure the thickness, stiffness, and echogenicity of the Rectus Femoris muscle as a non-invasive tool to diagnose and stratify the severity of ALD-induced sarcopenia. Sarcopenia represnets an new and pivotal paraemter in several medical condition. The paper therefore has some merit, however i have some concerns

Major Criticisms

The study lacks a direct comparison with the current gold standard for sarcopenia diagnosis, such as CT or MRI. While the study aims to validate ultrasound as a diagnostic tool, without comparison to established methods, the results are difficult to validate. Including a comparison with CT or MRI could strengthen the study's conclusions regarding the effectiveness of ultrasound. Please, if you cannot proved a control with gold standard diagnostic toools,  provied details about that even in form of discussion of previous findings in litteraturem or in your previous research.

The paper acknowledges the lack of a standardized protocol for ultrasound skeletal muscle measurements, specifically for muscle stiffness or echogenicity. This limitation can lead to variability in results and reduce the reproducibility of the study. Standardizing these measurements is crucial for the clinical application of the technique. Please specify

The paper discusses myosteatosis as a factor in muscle quality deterioration but does not provide a detailed analysis or differentiation between sarcopenia and myosteatosis in the context of ALD. A more in-depth exploration of how these conditions interact could provide valuable insights into disease mechanisms. It should be better discussed

minor Criticisms

The paper includes figures and tables, but the visual presentation could be improved for clarity. For instance, figures could benefit from more detailed captions that explain the significance of the data presented, and tables could include additional summary statistics for a more comprehensive overview.

Comments on the Quality of English Language

no comment

Author Response

Comment 1. The study lacks a direct comparison with the current gold standard for sarcopenia diagnosis, such as CT or MRI. While the study aims to validate ultrasound as a diagnostic tool, without comparison to established methods, the results are difficult to validate. Including a comparison with CT or MRI could strengthen the study's conclusions regarding the effectiveness of ultrasound. Please, if you cannot proved a control with gold standard diagnostic toools, provied details about that even in form of discussion of previous findings in litteraturem or in your previous research.

Response 1. As we mentioned, our study does not have a direct comparison of ultrasound measurement to gold-standard CT/MRI. Multiple validation studies have been conducted in this manner showing the reliability of ultrasound measurements when compared to CT scans [20, 24-26]. Muscle ultrasound measurements reliability as an alternative to CT scans is included in discussions. Appropriate references and comments were added in text (page 6)

Comment 2. The paper acknowledges the lack of a standardized protocol for ultrasound skeletal muscle measurements, specifically for muscle stiffness or echogenicity. This limitation can lead to variability in results and reduce the reproducibility of the study. Standardizing these measurements is crucial for the clinical application of the technique. Please specify

Response 2. As a novel diagnostic tool, standardization of muscle ultrasound assessment for the diagnosis of sarcopenia is an ongoing process. The difficulty arises from the heterogenicity of measurement protocols, including the transducer selection (linear/convex), minimal/maximum transducer compression, anatomical landmarks and also being operator-dependent. Recently the SARCopenia through UltraSound working group (SARCUS) proposed a protocol that was also used in our study. We added the comment and reference in the text (page 3 and 7).

Regarding muscle stiffness assessment through ARFI elastography there is little data as most studies used 2D share weave elastography. Regarding echogenicity we used a qualitative scale comparing the echogenicity of muscle to neighboring fat or bone tissue similar to qualitative liver steatosis assessment, since our ultrasound machine did not have the software for quantitative assessment. The protocol is discussed accordingly in text. 

Comment 3. The paper discusses myosteatosis as a factor in muscle quality deterioration but does not provide a detailed analysis or differentiation between sarcopenia and myosteatosis in the context of ALD. A more in-depth exploration of how these conditions interact could provide valuable insights into disease mechanisms. It should be better discussed.

Response 3. Sarcopenia and myosteatosis are two different entities. While sarcopenia can be quantitatively assessed using imagistic measurements, primarily muscle thickness, myosteatosis is a rather newly acknowledged concept that influences muscle functionality through adipose tissue deposit. Both adipose and fibrous tissue show as echointensity on ultrasound. Our study did not have a primary aim the assessment of myosteatosis since those studies should include a muscle biopsy. We merely pointed out that the prevalence of muscle echointensity was highest in alcoholic hepatitis that could be explained by either fat deposition or muscle fibrosis, even in patients with normal muscle thickness. This could represent a specific prognostic factor potentially unrelated to sarcopenia.  In order to better understand qualitative muscle modifications, their ultrasound expression and relation to disease severity, studies including muscle biopsy are needed.  

According comment was added in text (page 7)

Comment 4. The paper includes figures and tables, but the visual presentation could be improved for clarity. For instance, figures could benefit from more detailed captions that explain the significance of the data presented, and tables could include additional summary statistics for a more comprehensive overview.

Response 4. According modifications to figure 1 were made. Tables include all the parameters and statistical analysis relevant to the study.

Round 2

Reviewer 1 Report

Comments and Suggestions for Authors

In the section biological and clinical results there shouldn’t be the sentence” these results…” - it is kind of explanation which should be in discussion part (I mentioned it before). Authors didn’t explain the control group in the text (answering my question is not enough and it also should be explained why healthy people underwent this procedure). Only minor changes have been done with the text.

Comments on the Quality of English Language

Minor editorial changes

Author Response

Comment 1: In the section biological and clinical results there shouldn’t be the sentence” these results…” - it is kind of explanation which should be in discussion part (I mentioned it before). Authors didn’t explain the control group in the text (answering my question is not enough and it also should be explained why healthy people underwent this procedure). Only minor changes have been done with the text.

Response 1: We have made the necessary modifications. We added details about the control group on page 2 (Materials and Methods) as well as other comments. We explained that both 30sCST and ultrasound measurements were compared between groups.

Page 3: Control patients were assessed during a routine check-up that included clinical examination, blood samples and ultrasound assessment. US skeletal muscle findings were compared between groups and with 30sCST sarcopenia surrogate test.

All explanations from “Results” were moved and integrated in the “Discussion” section (page7)
